# Infection and vaccination status of COVID-19 among healthcare professionals in academic platform: Prevision vs. reality of Bangladesh context

Bilkis Banu[◉], Nasrin Akter[◉]*[◉], Sujana Haque Chowdhury[◉], Kazi Rakibul Islam[‡], Md. Tanzeerul Islam[‡], Sarder Mahmud Hossain

Department of Public Health, Northern University Bangladesh, Dhaka, Bangladesh

◉ These authors contributed equally to this work.
‡ KRI and MTI also contributed equally to this work.
* nasrin.ddc@gmail.com

## Abstract

COVID-19 posed the healthcare professionals at enormous risk during this pandemic era while vaccination was recommended as one of the effective preventive approaches. It was visualized that almost all health workforces would be under vaccination on a priority basis as they are the frontline fighters during this pandemic. This study was designed to explore the reality regarding infection and vaccination status of COVID-19 among healthcare professionals of Bangladesh. It was a web-based cross-sectional survey and conducted among 300 healthcare professionals available in the academic platform of Bangladesh. A multivariate logistic regression model was used for the analytical exploration. Adjusted and Unadjusted Odds Ratio (OR) with 95% confidence intervals (95% CI) were calculated for the specified setting indicators. A Chi-square test was used to observe the association. Ethical issues were maintained according to the guidance of the declaration of Helsinki. Study revealed that 41% of all respondents identified as COVID-19 positive whereas a significant number (18.3%) found as non-vaccinated due to registration issues as 52.70%, misconception regarding vaccination as 29.10%, and health-related issues as 18.20%. Respondents of more than 50 years of age found more significant on having positive infection rather than the younger age groups. Predictors for the non-vaccination guided that male respondents (COR/p = 3.49/0.01), allied health professionals, and respondents from the public organizations (p = 0.01) who were ≤29 (AOR/p = 4.45/0.01) years of age significantly identified as non-vaccinated. As the older female groups were found more infected and a significant number of health care professionals found as non-vaccinated, implementation of specific strategies and policies are needed to ensure the safety precautions and vaccination among such COVID-19 frontiers.

**Data Availability Statement:** Data cannot be made publicly available as it contains potentially identifying participant information. Ethical approval

was taken from the appropriate Ethical Review Committee of the Department of Public Health of Northern University Bangladesh. Strict confidentiality was maintained to collect and restore the data. Data is available on request from Md. Masum Parvez (Academic Officer, Department of Public Health, Northern University Bangladesh (NUB): parumasum83@gmail.com) to researchers who meet the criteria for access to confidential data.

**Funding:** There was no funding support in this project such as study design, data collection and analysis, decision to publish, or preparation of the manuscript. No authors received any specific funding for this work.

**Competing interests:** The authors have declared that no competing interests exist.

## Introduction

The reason behind corona virus disease is severe acute respiratory syndrome corona virus 2 (SARS-Cov-2). The disease then spread widely across the globe, causing a severe humanitarian and economic crisis, additionally to the burden on healthcare services. Vaccine hesitancy is one amongst the ten most serious threats to global health. As reported by WHO (2019), reasons behind refusal or unwillingness may be associated with inconvenience in accessing vaccines, complacency, or lack of trust [1]. There's no other strict way in lieu of vaccinations and it's the simplest method to regulate rapidly spreading infectious diseases. Waning public confidence in vaccines because of rumors and conspiracy theories may be a major challenge for public health experts and policymakers worldwide [2]. It will be a great strength to make sure the safety of health care workers by the proper implementation of vaccination program for an extended time. To complete vaccination status is incredibly much important for health care professionals under academic platform. These groups of scholars are at higher risk for of contacting and transmitting highly infectious diseases like COVID-19 [3].

Preventive measures face challenges in Bangladesh due to its high population density. Social distancing is difficult in most of the areas of the country, and that's why mitigation measures are highly challenging. The disease has an alarmingly escalating rate of spread which is significantly higher than many other infectious diseases. It did potentially lead to incapacitate the existing health facilities of the country. Similar scenario happened too even in developed countries with a rapidly increasing Novel coronavirus infection [4]. Health care professionals are usually in greater danger for increased exposure to infectious diseases compared to the common people within the community, and that they may act as vectors for transmission of this virus. Likewise, HCWs in hospital setting are also exposed repeatedly to the disease condition and may induce mutagenicity with development of vulnerable strain [5]. Consequently, this pandemic affected people of all strata within a short period of time particularly the frontline COVID workers on the first hand. Shortage of Doctors and Nurses is another trouble for Bangladesh [6]. A study from African region stated that only 27.7% HCWs would accept vaccine on the basis of availability [7]. COVID-19 posed a significant impact on most crucial sectors of Bangladesh including the economy, agriculture, education, and especially the health sector. As there is fewer data in this issue, our study will explore a new dimension with specific initiatives among the health care professionals in academic platforms. Identifying the present status of Covid-19 vaccination as well as speedy coverage among the health care professionals are further needed in the context of Bangladesh. The outcome of the study will pave suitable guidelines for future extensive research and way-outs for capacity building for full coverage of vaccination among the health care providers. The findings of the study may influence the government to undertake a sustainable program with the aim to stop the spread of the disease and to control the situation.

## Methods

### Study design

This cross-sectional study was carried out based on retrospective approach and the structured data were collected August 2021 to depict the infection and vaccination status of COVID-19 among healthcare professionals of Bangladesh.

### Study participants, sample size and sampling

A total 300 respondents were enrolled in this study who were involved as students of public health department of Northern University Bangladesh (NUB) and also serving as active healthcare professionals in different public and private organizations across the country. Department

of public health, NUB has been continuing evening and holiday/weakened based program of Master in Public Health (MPH) since 2006 for the physicians, nurses and allied graduate health professionals i.e. nutritionist, physiotherapist, laboratory technologist, Non-government Organization (NGO) specialist etc. Enrolment in the program are considered as part time academic involvement of the students, who as well as are engaged in their full-time professional work. All enrolled (400) students in summer-2021 semester (May to August 2021) of the MPH program of NUB were considered as participants in this study. Initially it was assumed that a potential standard sample size 384 would be taken by using the formula "$n = Z^2pq/d^2$" where Z (standard normal deviate) considered as 1.96; p (proportion of infected and vaccinated healthcare professionals) was unknown and was considered as 0.50 and margin of error was considered as 0.05. However, entirety number of MPH students directed to the declined respondent rate as 335 according to their response to the self-administered data collection instrument. After data cleaning and initial management final sample size was fixed at 300.

## Data collection

Data were gathered by self-administered, structured and anonymous online questionnaire. Due to the spread of the COVID-19 pandemic and the lockdown policy enforced within the country, a physical and paper-based questionnaire was not feasible. Thus, respondents were accessed through emails and social media platforms i.e. WhatsApp, and Face book Messenger concurrently. The web link of online survey was 'https://docs.google.com/forms/d/e/' which took only 3 to 4 minutes by the respondents to complete. The online web-based survey was administered in English language with the utmost support of the university authority.

## Ethical considerations

This study was approved by the Ethical Review Committee of the Department of Public Health of Northern University Bangladesh (NUB/DPH/EC/2021/05-a) and conformed to the Declaration of Helsinki. Participation of the respondents was anonymous and voluntary. Informed consent was sought from the respondents at the beginning of the survey and participants could withdraw from the survey at any time.

## Questionnaire design

A Google form was used to develop the online questionnaire. The questionnaire was pre-validated by two independent reviewers and pre-tested among 10 respondents. The responses from the pre-test were used to improve upon the quality of the questionnaire. The questionnaire comprised of several segments: (i) Identification of COVID-19 infection status among the health care professionals; (ii) Reveal of COVID-19 vaccination status including non-vaccination reasons; (iii) Demography of the healthcare professionals: age, gender, geographical location, occupation; (iv) Organizational information: organization type and duration of working experience.

## Data analysis

Quality of data was checked and analyzed employing the Statistical Package for the Social Sciences (SPSS) software. Study characteristics were subjected to descriptive statistics (frequency and proportions) to summarize the obtained data. To categorize the data of Age and Experience the cut off value was decided according to previous relevant published articles [8, 9]. A multivariable logistic regression analysis was performed followed by modeling procedure considering backward elimination process, including pre-specified confounders i.e. age, gender,

occupation, location, working experience and organization type. Adjusted Odds Ratios with 95% confidence intervals with respect to COVID-19 infection (test positive or test negative) and vaccination status (vaccinated or non-vaccinated) were calculated for the specified exposures.

## Results

### Participant's characteristics

A total of 300 respondents were included in this study with 77.7% female and a mean (±SD) age of 38.67 (±9) years. Among the respondents 39.3% (n = 118/300) belonged the age group of 40–49 years. Majority (69.3%, n = 208/300) of the respondents were Nurses and mostly from Dhaka district (65.7%, n = 197). In addition, more than half of the study subjects (57%, n = 171/300) had been serving in the private health care organizations, while 43% had been in government organizations. Moreover, study also revealed that more than half of the subjects (51.3%, n = 154/300) had more than 10 years of professional working experience (Table 1).

**Table 1. Characteristics of the respondents according to COVID-19 infection and vaccination status (n = 300).**

| Characteristics | COVID-19 infection status | | | | COVID-19 vaccination status | | | |
|---|---|---|---|---|---|---|---|---|
| | Number of participants, n (%) | Test Positive, n (%) | Test negative, n (%) | p-value (≤0.05) | Number of participants, n (%) | Vaccinated, n (%) | Non-vaccinated, n (%) | p-value (≤0.05) |
| **Age group (in years)** | | | | | | | | |
| ≤29 | 69 (23.0) | 19 (6.3) | 50 (16.7) | 0.01* | 69 (23.0) | 37 (12.3) | 32 (10.7) | 0.01* |
| 30–39 | 78 (26.0) | 33 (11.0) | 45 (15.0) | | 78 (26.0) | 64 (21.3) | 14 (4.7) | |
| 40–49 | 118 (39.3) | 49 (16.3) | 69 (23.0) | | 118 (39.3) | 114 (38.0) | 4 (1.3) | |
| >50 | 35 (11.7) | 22 (7.3) | 13 (4.3) | | 35 (11.7) | 30 (10.0) | 5 (1.7) | |
| **Gender** | | | | | | | | |
| Male | 67 (22.3) | 26 (8.7) | 41 (13.7) | 0.77 | 67 (22.3) | 47 (15.7) | 20 (6.7) | 0.01* |
| Female | 233 (77.7) | 97 (32.3) | 136 (45.3) | | 233 (77.7) | 198 (66.0) | 35 (11.7) | |
| **Profession** | | | | | | | | |
| Physicians | 44 (14.7) | 14 (4.7) | 30 (10) | 0.45 | 44 (14.7) | 32 (10.7) | 12 (4.0) | 0.01* |
| Nurses | 208 (69.3) | 87 (29.0) | 121 (40.3) | | 208 (69.3) | 183 (61.0) | 25 (8.3) | |
| Allied health professionals | 48 (16.0) | 22 (7.3) | 26 (8.7) | | 48 (16.0) | 30 (10.0) | 18 (6.0) | |
| **Experience** | | | | | | | | |
| ≤10 | 146 (48.7) | 51 (17.0) | 95 (31.7) | 0.04* | 146 (48.7) | 106 (35.3) | 40 (13.3) | 0.01* |
| >10 | 154 (51.3) | 72 (24.0) | 82 (27.3) | | 154 (51.3) | 139 (46.3) | 15 (5.0) | |
| **Type of Organization** | | | | | | | | |
| Private | 171 (57.0) | 74 (24.7) | 97 (32.3) | 0.45 | 171 (57.0) | 148 (49.3) | 23 (7.7) | 0.01* |
| Public | 129 (43.0) | 49 (16.3) | 80 (26.7) | | 129 (43.0) | 97 (32.3) | 32 (10.7) | |
| **Geographic Location** | | | | | | | | |
| Dhaka | 197 (65.7) | 84 (28.0) | 113 (37.7) | 0.57 | 197 (65.7) | 163 (54.3) | 34 (11.3) | 0.01* |
| Barisal | 24 (8.0) | 11 (3.7) | 13 (4.3) | | 24 (8.0) | 23 (7.7) | 1 (0.3) | |
| Chittagong | 46 (15.3) | 17 (5.7) | 29 (9.7) | | 46 (15.3) | 29 (9.7) | 17 (5.7) | |
| Khulna | 15 (5.0) | 3 (1.0) | 12 (4.0) | | 15 (5.0) | 15 (5.0) | 0 (0.0) | |
| Raj Shahi | 8 (2.7) | 3 (1.0) | 5 (1.7) | | 8 (2.7) | 8 (2.7) | 0 (0.0) | |
| Mymensingh | 10 (3.3) | 5 (1.7) | 5 (1.7) | | 10 (3.3) | 7 (2.3) | 3 (1.0) | |

Data are presented as frequency (n), percentage (%)

*Statistical significance at p value ≤0.05. Chi-square test was used to observe the association.

### COVID-19 infection status among the health professionals

Among the health care professionals, nearly half (41%, n = 123/300) were revealed as COVID-19 test positive whereas 59% were found negative (Fig 1).

### COVID-19 vaccination status among the health professionals

Most of the health care professionals (81.7%, n = 245/300) were found as vaccinated where a good number (18.3%, n = 55/300) did not take any vaccine (Fig 2). The reasons for not being vaccinated were Registration issues (52.70%), Misconception (29.10%) and Health related issues (18.20%) (Fig 2).

### Respondent's characteristics associated with the COVID-19 infection and vaccination status

Results of multivariate (cross table) analysis revealed that respondents' age (p = 0.01) and professional working experience (p = 0.04) is significantly associated with the COVID-19

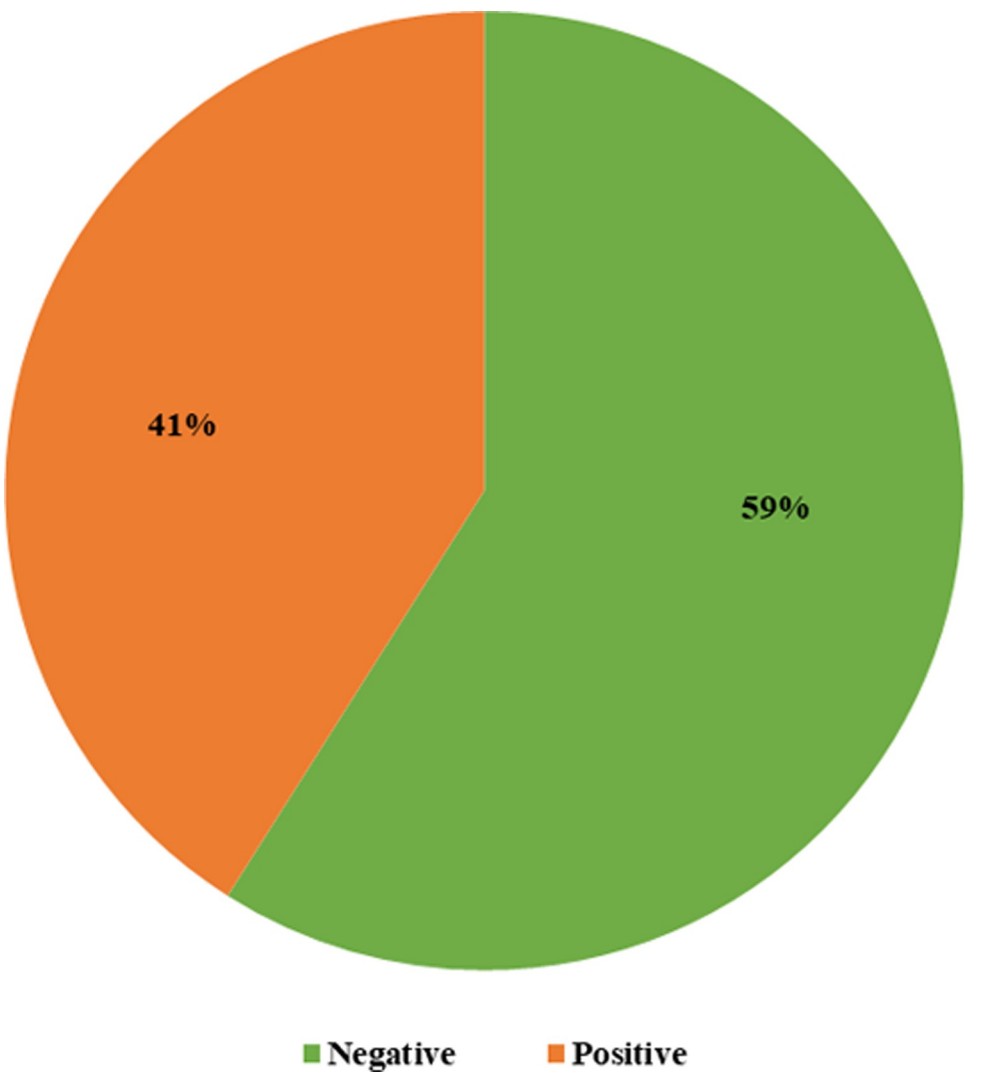

**Fig 1. This is the status of COVID-19 infection among the respondents (n = 300).**

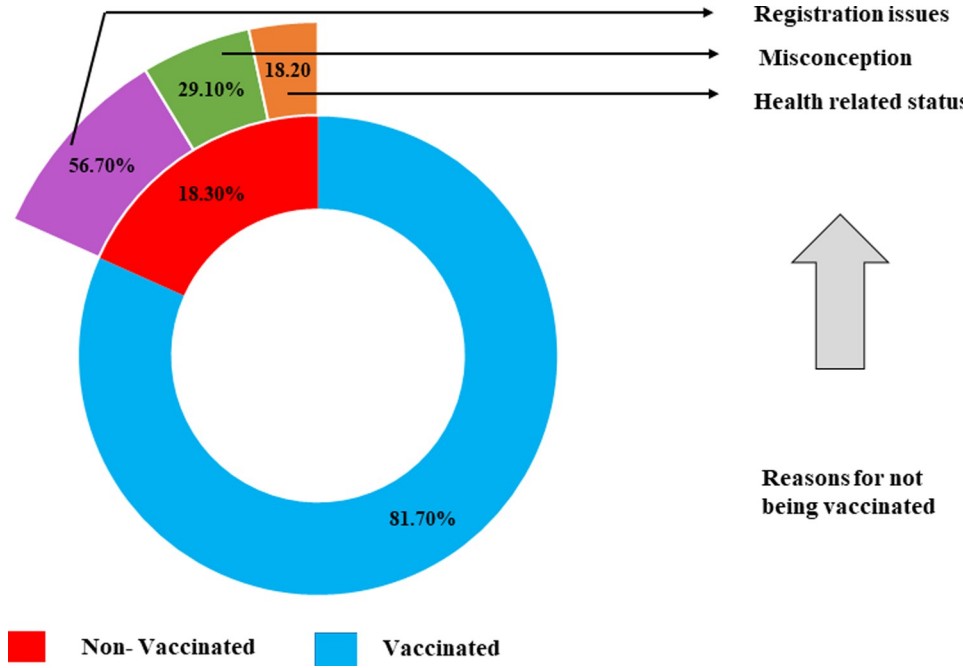

**Fig 2. This is the status of COVID-19 vaccination including reasons for non-vaccination among the respondents (n = 300).**

infection status. On the other hand, vaccination status among the respondents significantly influenced by the demographic characteristics like age, gender, profession, working experience, types of organization and geographic location (p = 0.01). Study also revealed that comparatively more COVID-19 infection found among the female who were nurse as their occupation whereas the group of allied health professionals found as the second largest group infected with COVID-19. In addition, COVID-19 infection predominantly found among the health care professionals who were from the Dhaka district. Furthermore, COVID-19 vaccination status revealed that 40–49 years age group and female subjects who were in nursing profession found more vaccinated, compared to others. HCWs who had more than 10 years of experience and from private organizations of Dhaka city were found more vaccinated than others (Table 1).

## Predictors regarding COVID-19 infection and vaccination status among the respondents

Multivariable logistic regression analysis was done to identify the predictors. A backward step-by-step binary logistic regression (simple and multiple) was performed after the placement of all the significant predictors (p<0.05) in final model. It was revealed from the adjusted model that respondents aged more than 50 years were more COVID positive rather than the younger age groups ($\leq$29 years, COR/p = 0.23/0.01, AOR = 0.22/0.01; 30–39 years, COR/p = 0.43/0.04 & AOR/p = 0.43/0.04; 40–49 years, COR/p = 0.42/0.03 & AOR/p = 0.42/0.03). On the other hand, predictors regarding COVID-19 vaccination status revealed that respondents who were $\leq$29 (AOR/p = 4.45/0.01) and >50 years of age found significantly reluctant to be vaccinated. In addition, male (COR/p = 3.49/0.01) respondents, allied health professionals and respondents from the public organizations were (p = 0.01) significantly identified as non-vaccinated compared to corresponding groups (Table 2).

**Table 2. Identified predictors associated with the COVID-19 infection and vaccination status (n = 300).**

| Characteristics | COVID-19 infection status | | | | COVID-19 vaccination status | | | |
| --- | --- | --- | --- | --- | --- | --- | --- | --- |
| | Test negative vs test positive counter | | | | Non-Vaccinated vs vaccinated counter | | | |
| | Un-adjusted OR (95% CI) | p-value | Adjusted OR (95% CI) | p-value | Un-adjusted OR (95% CI) | p-value | Adjusted OR (95% CI) | p-value |
| **Age group (in years)** | | | | | | | | |
| ≤29 | 0.23 (0.09–0.53) | 0.01* | 0.22 (0.09–0.53) | 0.01* | 5.19 (1.80–14.95) | 0.01* | 4.45 (1.51–13.08) | 0.01* |
| 30–39 | 0.43 (0.19–0.98) | 0.04* | 0.43 (0.19–0.98) | 0.04* | 1.31 (0.43–3.98) | 0.63 | 1.09 (0.35–3.39) | 0.88 |
| 40–49 | 0.42 (0.19–0.91) | 0.03* | 0.42 (0.19–0.91) | 0.03* | 0.21 (0.05–0.83) | 0.03* | 0.22 (0.06–0.89) | 0.03* |
| >50 | Reference | | | | Reference | | | |
| **Gender** | | | | | | | | |
| Male | 0.89 (0.51–1.55) | 0.68 | — | — | 2.41 (1.28–4.54) | 0.01* | — | — |
| Female | Reference | | | | Reference | | | |
| **Profession** | | | | | | | | |
| Physicians | 0.55 (0.24–1.29) | 0.17 | — | — | 0.63 (0.26–1.51) | 0.29 | 0.56 (0.21–1.47) | 0.24 |
| Nurses | 0.85 (0.46–1.59) | 0.61 | — | — | 0.23 (0.11–0.47) | 0.01* | 0.34 (0.15–0.76) | 0.01* |
| Allied health professionals | Reference | | | | Reference | | | |
| **Experience** | | | | | | | | |
| ≤10 | 0.61 (0.38–0.97) | 0.04* | — | — | 3.49 (1.84–6.67) | 0.01* | — | — |
| >10 | Reference | | | | Reference | | | |
| **Type of Organization** | | | | | | | | |
| Private | 1.25 (0.78–1.99) | 0.36 | — | — | 0.47 (0.26–0.85) | 0.01* | — | — |
| Public | Reference | | | | Reference | | | |
| **Geographic Location** | | | | | | | | |
| Dhaka | 0.74 (0.21–2.65) | 0.65 | — | 0.03* | 0.49 (0.12–1.98) | 0.31 | — | — |
| Barisal | 0.85 (0.19–3.71) | 0.83 | — | 0.08 | 0.10 (0.01–1.14) | 0.06 | — | — |
| Chittagong | 0.59 (0.15–2.32) | 0.45 | — | 0.04* | 1.36 (0.31–6.01) | 0.68 | — | — |
| Khulna | 0.13 (0.04–1.47) | 0.13 | — | — | — | — | — | — |
| Raj Shahi | 0.59 (0.09–3.98) | 0.59 | — | — | — | — | — | — |
| Mymensingh | Reference | | | | Reference | | | |

Logistic Regression Analysis was used to identify the predictors

* Statistical significance at p value ≤0.05.

## Discussions

Availability and efficacy of the COVID-19 vaccine are vivacious for a successfully control of pandemic. Our study identifies COVID-19 status and vaccination and its predictor for non-vaccination status of among health workers in Bangladesh. Majority of the participants were nurses (69.3%) followed by Allied health professionals (16%), and physicians (14.7%). Highlighting upon the test result of COVID-19 41% reported positive test result among total respondents, while another cohort study in New Jersey, USA reported that only 5.0% of all health care providers were tested positive in university hospital among whom majority (62.5%) were nurses and the positive tests increased across the two weeks of cohort recruitment. To minimize the prevalence they recommended continuous follow-up, monitoring infection rates and examine risk factors for transmission as the possible way outs need to come in action [10]. Compare to that scenario, as infection rate is much higher in Bangladesh, it is pivotal issue for the rapid surveillance. In addition, monitoring and screening system of

COVID-19 infection with risk factor identification needs to be emphasized among the health care providers of the country.

Moving upon vaccination status it was observed that 81.7% were found as vaccinated where a significant amount 18.3% were non-vaccinated. However, it was expected that as frontiers vaccination program will cover all the health care professionals as priority according to the registration guideline of vaccination in Bangladesh [11]. As the scientific evidence regarding vaccination status among health care providers is not available in the intellectual platform, the outcome of this study might give a new outlook for the health wellbeing of this vital group in our community. From nurses' group highest (87.9%) number of participants reported a complete vaccination status compare to physician group, whereas lowest (10%) was reported from alight group of respondents. There are a very few studies available indicating the COVID-19 status and vaccination among healthcare professionals as frontline workers in Bangladesh. While identifying the reasons for not being vaccinated, it was found that "registration issues" like waiting for confirmation massage was reported highest (52.70%) and list (18.20%) reported fact was having health issues like pregnancy. Furthermore, misconception regarding vaccination was observed among (29.1%) healthcare workers (HCWs). This study findings coincide with another study in Bangladesh says that HCWs think the vaccine might not be safe or effective due to emergency authorization [12]. Similar in a study from Ghana, finding was observed that healthcare workers in Ghana got misconception regarding safety and adverse side effects of the vaccines, which identified as main reasons why health care workers would decline uptake of COVID-19 vaccines [13]. Association between socio demographic factors and COVID status, it was found that age and working experience of respondents had a positive relation. Respondent having working experience more than 10 years reported highest (51.3%) positive test result of COVID-19. Having good management skill with increased workload or less availability of PPE may act as driving impetus behind more positive test result and handling more COVID patient. Furthermore, age group between 40–49 years reported maximum infectivity history compare to other age group, whereas age group less than 29 reported minimum (23%) infectivity history. This result part agrees with a study in Bangladesh reveal that HCWs whose age is under 20 years are more likely to not be infected by COVID-19 than those above that age group [14].

Highlighting upon the association between vaccination status and socio demographic factors, every factor like age, gender, profession, experience, organization type found to have significant association with vaccination. In this study female, nurses in age group 40–49, serving in a private organization with a working experience >10 reported complete vaccination status. This study finding unfortunately contradict with a study finding says that female found to be a contributing factor to the low vaccine acceptance rate in Egypt [15].

Concerning upon the predictor respondents aged more than 50 years found significantly reporting of having positive infection history rather than the younger age groups. Mostly similar to this finding coincide to a study identifying highest infection and deaths reported among HCWs aged over 70 years per 100 infections [16]. Predictors concerning age related to COVID-19 vaccination status it was found that respondents age group ≤29 identified as significant predictor for non-vaccination. Extreme age group with undiversified working experience and knowledge may act as main reason for this reluctance. Very few studies found supporting this study result. In addition, male respondents were more likely to be non-vaccinated. This study finding contradict with many of article, which says that female respondents were more reluctant toward the vaccine where safety and efficacy concerns of vaccine were the significant predictors of vaccine hesitancy [17]. Allied health professionals and respondents from the public organizations were found to be significant predictor for non-vaccinated in this

pandemic. Prolong registration procedure or working hour may work behind this phenomenon.

Although our students representing a large number of health care professionals throughout Bangladesh but we could not develop the sampling frame from all the health care facilities or organizations from the whole country due to lack of resources, funding and time constrains. Therefore, we did this formative study visualizing the initial situation of vaccination status and the associated risk factors among the health care professionals in Bangladesh. The unique outcome of this study is the strength of the preventive approach of COVID-19 infection among the health care professionals which might be the mounting for further scientific initiatives. Thus, the further large-scale survey is needed to generalize the conclusion as well as planning for the specific intervention in priority basis.

## Conclusions

A significant diversity was found in this study concerning Covid infection and vaccination status among different categories of health professionals in the academic settings. Nearly half of respondents had COVID-19 positive infection whereas 18.3% of respondents were found as non-vaccinated. It was an alarming health issue in a serious ongoing pandemic situation like novel corona virus infection. Although the female nurses' group was more infected as a front-line healthcare provider in spite of their higher vaccination status. Less response on vaccination was found among the allied and younger age group health professionals. The caregiver with less professional experience (<10 years) were found as more positive and non-vaccinated. Registration issues and misconceptions about vaccines were branded as first and second most leading cause behind the non-vaccination status. Thus, clarity-based scrutinization is required on infection control measures for such front-line health workers. Proper approaches are needed to ensure a successful vaccination program among the health professionals to inspire a voluntary successful vaccination. Health awareness programs highlighting the benefits of vaccination through behavior change communication are needed to aware such populations to ensure a safer workplace.

## Supporting information

**S1 File.**
(DOCX)

**S2 File.**
(SAV)

**S1 Data.**
(CSV)

## Acknowledgments

We strongly acknowledge the study participants and the authority of the study place.

## Author Contributions

**Conceptualization:** Bilkis Banu, Nasrin Akter, Sujana Haque Chowdhury, Kazi Rakibul Islam, Md. Tanzeerul Islam, Sarder Mahmud Hossain.

**Data curation:** Bilkis Banu, Nasrin Akter, Sujana Haque Chowdhury, Kazi Rakibul Islam, Md. Tanzeerul Islam, Sarder Mahmud Hossain.

**Formal analysis:** Bilkis Banu, Nasrin Akter, Sujana Haque Chowdhury, Kazi Rakibul Islam, Md. Tanzeerul Islam, Sarder Mahmud Hossain.

**Funding acquisition:** Bilkis Banu, Nasrin Akter, Sujana Haque Chowdhury, Kazi Rakibul Islam, Md. Tanzeerul Islam, Sarder Mahmud Hossain.

**Investigation:** Bilkis Banu, Nasrin Akter, Sujana Haque Chowdhury, Kazi Rakibul Islam, Md. Tanzeerul Islam, Sarder Mahmud Hossain.

**Methodology:** Bilkis Banu, Nasrin Akter, Sujana Haque Chowdhury, Kazi Rakibul Islam, Md. Tanzeerul Islam, Sarder Mahmud Hossain.

**Project administration:** Bilkis Banu, Nasrin Akter, Sujana Haque Chowdhury, Kazi Rakibul Islam, Md. Tanzeerul Islam, Sarder Mahmud Hossain.

**Resources:** Bilkis Banu, Nasrin Akter, Sujana Haque Chowdhury, Kazi Rakibul Islam, Md. Tanzeerul Islam, Sarder Mahmud Hossain.

**Software:** Bilkis Banu, Nasrin Akter, Sujana Haque Chowdhury, Kazi Rakibul Islam, Md. Tanzeerul Islam, Sarder Mahmud Hossain.

**Supervision:** Bilkis Banu, Nasrin Akter, Sujana Haque Chowdhury, Kazi Rakibul Islam, Md. Tanzeerul Islam, Sarder Mahmud Hossain.

**Validation:** Bilkis Banu, Nasrin Akter, Sujana Haque Chowdhury, Kazi Rakibul Islam, Md. Tanzeerul Islam, Sarder Mahmud Hossain.

**Visualization:** Bilkis Banu, Nasrin Akter, Sujana Haque Chowdhury, Kazi Rakibul Islam, Md. Tanzeerul Islam, Sarder Mahmud Hossain.

**Writing – original draft:** Bilkis Banu, Nasrin Akter, Sujana Haque Chowdhury, Kazi Rakibul Islam, Md. Tanzeerul Islam, Sarder Mahmud Hossain.

**Writing – review & editing:** Bilkis Banu, Nasrin Akter, Sujana Haque Chowdhury, Kazi Rakibul Islam, Md. Tanzeerul Islam, Sarder Mahmud Hossain.

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
