## [Decision Letter · Decision Letter 0]

5 Nov 2021

PONE-D-21-30587Infection and vaccination status of COVID-19 among healthcare professionals in academic platform: prevision vs. reality of Bangladesh contextPLOS ONE

Dear Dr. Akter,

Thank you for submitting your manuscript to PLOS ONE. After careful consideration, we feel that it has merit but does not fully meet PLOS ONE’s publication criteria as it currently stands. Therefore, we invite you to submit a revised version of the manuscript that addresses the points raised during the review process.

We look forward to receiving your revised manuscript.

Kind regards,

Khin Thet Wai, MBBS, MPH, MA

Academic Editor

PLOS ONE

Journal Requirements:

2. PLOS ONE does not copy edit accepted manuscripts (https://journals.plos.org/plosone/s/criteria-for-publication#loc-5). To that effect, please ensure that your submission is free of typos and grammatical errors.

*Please include additional information regarding the survey or questionnaire used in the study and ensure that you have provided sufficient details that others could replicate the analyses. For instance, if you developed a questionnaire as part of this study and it is not under a copyright more restrictive than CC-BY, please include a copy, in both the original language and English, as Supporting Information.

 [No funding support. The funders had no role in study design, data collection and analysis, decision to publish, or preparation of the manuscript.]

5. Please include your tables as part of your main manuscript and remove the individual files. Please note that supplementary tables (should remain/ be uploaded) as separate "supporting information" files.

Additional Editor Comments:

This is the important manuscript that requires an improvement.

1. Authors need to modify the data analysis, interpretation of results and to strengthen the discussion section.

2. There are grammatical errors throughout and English language editing is deemed necessary.

Reviewers' comments:

Reviewer's Responses to Questions

**Comments to the Author**

1. Is the manuscript technically sound, and do the data support the conclusions?

Reviewer #1: Partly

Reviewer #2: Yes

2. Has the statistical analysis been performed appropriately and rigorously? 

Reviewer #1: Yes

Reviewer #2: Yes

3. Have the authors made all data underlying the findings in their manuscript fully available?

Reviewer #1: Yes

Reviewer #2: Yes

4. Is the manuscript presented in an intelligible fashion and written in standard English?

Reviewer #1: Yes

Reviewer #2: Yes

5. Review Comments to the Author

Reviewer #1: This study is interesting and important study for Bangladesh in fighting against COVID-19 outbreak. But, the manuscript still needs to improve the data analysis, presentation of the findings and implication of the study.

Background

- Need to describe the justification of the study clearly

- Study design with retrospective and structure: What does it mean?

- What is your study setting? Which responsibility did your participants take in which region? Are they still students or already graduated?

- In data analysis, need to describe how did you consider your final multivariable logistic model for each outcome variable

- When I check in the table 2, there are only 1 or 2 variables for AOR? Is it final model? Please explain.

Result session:

- In my opinion, 41% is not nearly half.

- Fig 1 and 2 should be pie chart. Figure 1 and 2 can be only one figure saying Fig 1.1 and 1.2. Fig 3 can also be combined to figure 2. Vaccination status and reasons for not having vaccines

- In my opinion, Table 1 and 2 should not be supplementary tables. Should describe as main table.

- Description of the results should be in text. Should not under the figure as picture.

- Regarding the result of preditors, I would like to suggest to describe the results from multivariable logistic regression with AOR.

- Did you add vaccine status as one of the predictor for COVID-19 infection?

Discussion

- Please discuss about 41% has infected. Please compare with other studies and what is the implication of your study? How will you apply?

- Please discuss about 81.7% has infected. Please compare with other studies and what is the implication of your study? How will you apply?

- Please describe the strength and limitation of your study.

- Please clearly describe the implication from your study.

- Please mention how you can make generalizability from your study population and sample size. Can we conclude this study represents all the health professional?

Reviewer #2: overall the manuscript technically good and data analysis ok and overall well written except few grammatic errors reasonably current pandaemic this paper will open up most of the countries still campaigning for vaccination

ok for acceptance

6. PLOS authors have the option to publish the peer review history of their article (what does this mean?). If published, this will include your full peer review and any attached files.

Reviewer #1: **Yes: **Thae Maung Maung

Reviewer #2: **Yes: **Kandamaran Krishnamurthy

---

## [Author Response · Author response to Decision Letter 0]

6 Dec 2021

Reviewer's Comment Response

Background

1. Need to describe the justification of the study clearly Modified in line 71-86

2. Study design with retrospective and structure: What does it mean? Modified in line 91-92

3. What is your study setting? Which responsibility did your participants take in which region? Are they still students or already graduated? Already Mentioned in line 95-96 and 100-101

4. In data analysis, need to describe how did you consider your final multivariable logistic model for each outcome variable Modified in line 133-134

5. When I check in the table 2, there are only 1 or 2 variables for AOR? Is its final model? Please explain. Yes, it is the final model. We did the final model by backward elimination of the study variables.

Result 

1. In my opinion, 41% is not nearly half. Modified in line 149-50

2. Fig 1 and 2 should be pie chart. Figure 1 and 2 can be only one figure saying Fig 1.1 and 1.2. Fig 3 can also be combined to figure 2. Vaccination status and reasons for not having vaccines. Modified as fig 1 and fig 2. We have merged fig 2 & 3 into new fig 2.

3. In my opinion, Table 1 and 2 should not be supplementary tables. Should describe as main table. Added in the result section

4. Description of the results should be in text. Should not under the figure as picture. Modified the figures

5. Regarding the result of predators, I would like to suggest to describe the results from multivariable logistic regression with AOR. Already mentioned in result section in line 180-186

6. Did you add vaccine status as one of the predictors for COVID-19 infection? No, we did not find any association between vaccination status and infection status.

Discussion

1. Please discuss about 41% has infected. Please compare with other studies and what is the implication of your study? How will you apply? Modified in lines 197-206

2. Please discuss about 81.7% has infected. Please compare with other studies and what is the implication of your study? How will you apply? Modified in lines 207-219

3. Please describe the strength and limitation of your study. Added in lines 257-265

4. Please clearly describe the implication from your study. 

5. Please mention how you can make generalizability from your study population and sample size. Can we conclude this study represents all the health professional? 

Editor's Comment 

Authors need to modify the data analysis, interpretation of results and to strengthen the discussion section. Addressed in the track changed manuscript file

There are grammatical errors throughout and English language editing is deemed necessary. Corrected and modified

---

## [Decision Letter · Decision Letter 1]

5 Jan 2022

PONE-D-21-30587R1Infection and vaccination status of COVID-19 among healthcare professionals in academic platform: prevision vs. reality of Bangladesh contextPLOS ONE

Dear Dr. Akter,

Thank you for submitting your manuscript to PLOS ONE. After careful consideration, we feel that it has merit but does not fully meet PLOS ONE’s publication criteria as it currently stands. Therefore, we invite you to submit a revised version of the manuscript that addresses the points raised during the review process.

We look forward to receiving your revised manuscript.

Kind regards,

Khin Thet Wai, MBBS, MPH, MA

Academic Editor

PLOS ONE

Journal Requirements:

Additional Editor Comments (if provided):

Minor grammatical errors throughout the manuscript require correction.

Reviewers' comments:

Reviewer's Responses to Questions

**Comments to the Author**

1. If the authors have adequately addressed your comments raised in a previous round of review and you feel that this manuscript is now acceptable for publication, you may indicate that here to bypass the “Comments to the Author” section, enter your conflict of interest statement in the “Confidential to Editor” section, and submit your "Accept" recommendation.

Reviewer #1: All comments have been addressed

2. Is the manuscript technically sound, and do the data support the conclusions?

Reviewer #1: Yes

3. Has the statistical analysis been performed appropriately and rigorously? 

Reviewer #1: No

4. Have the authors made all data underlying the findings in their manuscript fully available?

Reviewer #1: Yes

5. Is the manuscript presented in an intelligible fashion and written in standard English?

Reviewer #1: Yes

6. Review Comments to the Author

Reviewer #1: Reviewer’s comments_R1

- What does it mean “retrospective approach” in line 83-84.

This cross-sectional study was carried out based on retrospective approach and the structured data were collected

- Need to mention more detail about the student participants to get clear presentation for readers. Not enough described in line 95-96 and 100-101.

- What is the total number of students in in summer-2021 semester (May to August 2021) of the MPH program of NUB? Are public health students are part-time students? Are they also working?

- The sample size said 384 but the study enrolled 300 students? What is your explanation for that?

- In data analysis, AOR should be used in stead of OR in multivariable logistic regression.

7. PLOS authors have the option to publish the peer review history of their article (what does this mean?). If published, this will include your full peer review and any attached files.

Reviewer #1: **Yes: **Yes

---

## [Author Response · Author response to Decision Letter 1]

11 Jan 2022

Reviewer's Comment

1. What does it mean “retrospective approach” in line 83-84. 

Response: As the data was collected based on past history of their COVOD-19 infection and vaccination as we mentioned in the method. 

2. Need to mention more detail about the student participants to get clear presentation for readers. Not enough described in line 95-96 and 100-101. 

Response: Already Mentioned in line 95-96 and 100-101

3. What is the total number of students in in summer-2021 semester (May to August 2021) of the MPH program of NUB? Are public health students being part-time students? Are they also working? 

Response: Modified in line 91-95

4. The sample size said 384 but the study enrolled 300 students? What is your explanation for that? 

Response: Already Mentioned in line 101-103 

5. In data analysis, AOR should be used instead of OR in multivariable logistic regression. 

Response: Modified in line 133

Editor's Comment 

Minor grammatical errors throughout the manuscript require correction. 

Response: Already addressed

---

## [Editor Report · Decision Letter 2]

12 Jan 2022

Infection and vaccination status of COVID-19 among healthcare professionals in academic platform: prevision vs. reality of Bangladesh context

PONE-D-21-30587R2

Dear Dr. Akter,

We’re pleased to inform you that your manuscript has been judged scientifically suitable for publication and will be formally accepted for publication once it meets all outstanding technical requirements.

Kind regards,

Khin Thet Wai, MBBS, MPH, MA

Academic Editor

PLOS ONE
---

## [Editor Report · Acceptance letter]

8 Feb 2022

PONE-D-21-30587R2 

Infection and vaccination status of COVID-19 among healthcare professionals in academic platform: prevision vs. reality of Bangladesh context 

Dear Dr. Akter:

I'm pleased to inform you that your manuscript has been deemed suitable for publication in PLOS ONE. Congratulations! Your manuscript is now with our production department. 

Kind regards, 

on behalf of

Dr. Khin Thet Wai 

Academic Editor

PLOS ONE